# Use of Smartphone Applications in English Language Learning—A Challenge for Foreign Language Education

**Jaroslav Kacetl and Blanka Klímová \*** 

Department of Applied Linguistics, Faculty of Informatics and Management, University of Hradec Králové, Rokitanského 62, 500 03 Hradec Králové, Czech Republic

\*   Correspondence: blanka.klimova@uhk.cz

**Abstract:** At present, hardly any younger person can imagine life without mobile technologies. They use them on a daily basis, including in language learning. Such learning supported with mobile devices is called mobile learning, which seems beneficial especially thanks to the unique features of mobile applications (e.g., interactivity, ubiquity, and portability) and teachers' encouragement and feedback. The purpose of this review study is to explore original, peer-reviewed English studies from 2015 to April 2019 and to determine whether mobile applications used in the learning of English as a foreign language are beneficial and/or effective. The methods are based on a literature review of available sources found on the research topic in two acknowledged databases: Web of Science and Scopus. Altogether, 16 original journal studies on the research topic were detected. The results reveal that mobile learning is becoming a salient feature of education as it is a great opportunity for foreign language learning. Its key benefits are as follows: the enhancement of the learner's cognitive capacity, the learner's motivation to study in both formal and informal settings, the learner's autonomy and confidence, as well as the promotion of personalized learning, helping low-achieving students to reach their study goals. Although mobile learning seems to be effective overall, it is desirable to design, plan and implement it with caution, according to students' needs, and to deliver multiple language skills in authentic learning environments.

**Keywords:** mobile apps; mobile learning; English learning; use; benefits

## 1. Introduction

Nowadays, mobile technologies and mobile applications (apps) are becoming an indispensable part of learning, including foreign language learning [1]. This recent methodology of their use is called mobile learning (m-learning). M-learning further expands e-learning by promoting independent and active learning and by turning educational institutions into 24/7, no-barrier learning centers [2]. In a similar vein, Klimova [3] speaks of Mobile Assisted Language Learning (MALL) as a new subdivision of Computer Assisted Language Learning (CALL). Leis et al. [4] even suggest a new acronym for Smartphone Assisted Language Learning, SPALL, as the smartphone offers capabilities far beyond the traditional mobile phone.

The key features of m-learning, such as personalized learning, independent on time and place, collaboration with peers and teachers in both formal and informal settings, ubiquity and interactivity of mobile devices, make m-learning efficient [1,3].

Furthermore, research in MALL shows that using mobile phones and their apps seems to be beneficial for foreign language learning, especially thanks to their unique features (e.g., interactivity, ubiquity, or portability) and teachers' encouragement and feedback [5–8].

However, Klimova [3,5] mentions several pitfalls of MALL; namely, students' potential lack of attention caused by mobile phone multi-tasking, the lack of apps suitable for English for Specific Purposes (ESP) and at various levels of proficiency. Among the downsides are also problems with Internet access and connection, a small screen size, or a lack of face-to-face contact. In addition, Andersen [9] reports that the feedback function in the mobile apps is limited. In relation to foreign language learning apps, Heil et al. [10] state that most of the apps are decontextualized, i.e., they concentrate on individual words rather than on authentic speech production, including all four language skills (speaking, writing, listening and reading). They also emphasize the implementation of so-called adaptive learning, which tries to meet the unique needs of an individual through just-in-time feedback, pathways, and resources (rather than providing a one-size-fits-all learning experience) [11]. Furthermore, research shows that MALL is especially effective in vocabulary learning [3,5,12] because vocabulary can be split into smaller segments, which is suitable for designing content in smartphones.

Currently, the practice in the use of mobile apps in language learning is that they are mostly used as a support in language acquisition. Therefore, the blended learning (BL) approach (a combination of face-to-face instruction and online learning) is mostly implemented in relation to their use [12]. Overall, the BL approach appears to be more effective than the use of only traditional instruction. In addition, the BL approach is especially suitable for distant students, who due to their work commitments cannot be involved in full-time English language study [12].

The purpose of this review study is to explore original, peer-reviewed English studies from 2015 to April 2019 and to determine whether mobile apps used in the learning of English as a foreign language are beneficial and/or effective. Thus, the research question is as follows:

*Is the use of mobile apps beneficial and/or effective, in the learning of English as a foreign language?*
*(If so, why, in what ways, and how?)*

## 2. Methods

The methods are based on a literature review of available sources found on the research topic in two acknowledged databases: Web of Science and Scopus. The search period was conducted for studies published between January 2015 and April 2019, since several review studies [1,3,13–15] on this topic had been published before. The searched collocated keywords were as follows: effectiveness AND mobile apps AND English learning, effect AND mobile learning AND English learning, effectiveness AND use of mobile applications AND English language learning. The keywords were combined and integrated in database and journal searches. The terms used were searched using 'AND' to combine the keywords listed and using 'OR' to remove search duplication where possible. A backward search was also conducted, i.e., references of retrieved articles were assessed for relevant articles that authors' searches may have missed.

From the database/journal searches, 387 titles/abstracts were identified on the basis of the keywords. More studies were identified in the database Web of Science (248 studies). In SCOPUS, it was only 139 studies. Another five articles were identified from other sources, usually from references of the already detected articles. In addition, the authors performed a more specific search for only the peer-reviewed original journal articles, thus excluding conference articles and review articles. This generated altogether 190 original studies. After removing duplicates and titles/abstracts unrelated to the research topic, 115 English-written studies remained. Of these, only 58 articles were relevant for the research topic. These studies were investigated in full, and they were considered against the following inclusion and exclusion criteria. The inclusion criteria were as follows:

- The period of the publishing of the article was limited from 1 January 2015 up to 30 April 2019;
- Only reviewed full-text studies in scientific journals in English were included;
- Only experimental/quasi-experimental studies were included;
- The primary outcome focused on the association of the effectiveness of the use of smartphone apps in the learning of English as a foreign language.

The exclusion criteria were as follows:

- Conference papers, e.g., [7,8,16], review studies, e.g., [1,3], and original papers not focusing on smartphone apps for the learning of English as a second language, e.g., [17,18], were excluded.

Based on these criteria, 16 studies were eventually involved into the final analysis. Figure 1 below illustrates the selection procedure.

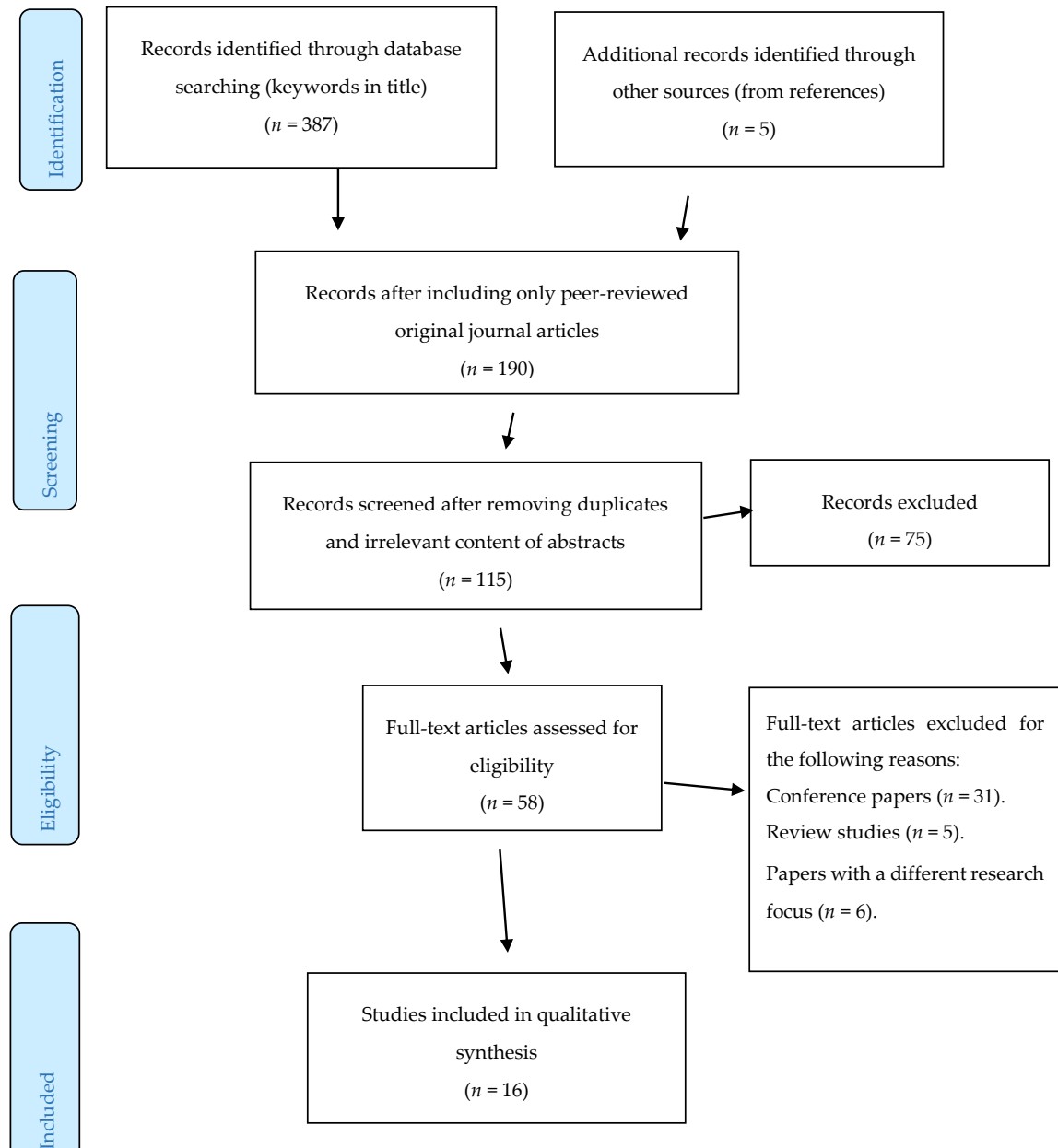

**Figure 1.** An overview of the selection procedure.

## 3. Results

Altogether, 16 original peer-reviewed journal articles on the research topic were detected. They originated in eleven different countries, namely, China, the Czech Republic, Iran, Japan, Lebanon, Russia, Saudi Arabia, Spain, Sri Lanka, Taiwan, and Turkey. However, the highest number of articles (3) were written in Iran and Taiwan, followed by China (2).

The majority of the 16 original reviewed texts agree that mobile learning, or m-learning, is becoming a salient feature of education (e.g., [19]), as it is a great opportunity and an immense step forward (e.g., [2,20,21]) and should therefore be supported (e.g., [22]), albeit with caution (e.g., [23,24]) and only as a supporting tool (e.g., [19]).

It is noteworthy that students usually report that they like m-learning [25]. This view is supported by other authors who also maintain that using mobile devices like smartphones and tablets in education is gladly accepted by learners (e.g., [2,21,25,26]). M-learning is sometimes used in order to help low-achieving students and motivate them (and others) to spend more time studying outside the classroom (e.g., [22]).

Teachers may use either some already established social networking platforms (WhatsApp, WeChat, Telegram, Line), or special applications for learning English (Fun Dubbing), or they may follow guidelines in creating tailor-made language learning apps (e.g., [19]).

As regards language skills practiced by means of m-learning, some research aimed at specific ones such as writing (e.g., [20,27]), speaking [28], vocabulary [3,21,23,25], listening [28], and reading [29].

The key findings of the selected studies are described in more detail below. The research was conducted mostly among secondary school and university students.

The mobile Internet has, according to Jin and Yan [30], distinct features, namely, convenience, portability, immediacy, orientation, accuracy, and sensitivity, which makes it much different from the desktop Internet. Jin and Yan [30] maintain that the effect of mobile learning is good as all students improve. On the other hand, some teachers as well as parents still resist to mobile learning as they do not understand it. Moreover, the teacher needs to invest a lot of time, and students lack self-confidence to ask questions [30].

Jamaldeen et al. [19] consider mobile learning to be one of the major developing areas in education. They tested a mobile-based learning application, and they claim that the users showed positive attitudes towards m-learning and found it useful. On the other hand, their findings suggest that m-learning would be more effective as a supporting medium of learning rather than as the primary medium (Jamaldeen et al., 2018).

Çelik and Yavuz [23] maintain that mobile apps help integrate smartphones into radically changing education, which is now more individualized, ubiquitous, learner-centered, and even uncontrolled. The authors studied the effectiveness of mobile apps in vocabulary instruction, both contextual and literal. They conclude that mobile apps are effective in language learning, but they warn that their implementation must be done in a guided and controlled way as some apps are not designed by experts.

Kuimova et al. [2] view m-learning as an important step forward and a valuable support to traditional learning. Their paper looks into the benefits and challenges of m-learning. According to these authors, m-learning enhances cognitive activity, encourages the learner's independence, helps individualize learning, and increases the learner's motivation. On the other hand, among its downsides are, e.g., small screens, potential external interference, a highly addictive as well as distractive nature, or the fact that some teachers are difficult to convince about m-learning's potential. Kuimova et al. [2] conducted research into using WhatsApp for learning English, and they conclude that mobile phones can be used for learning as students usually take a positive stance with regard to m-learning.

Awada [20] also looked into the effectiveness of WhatsApp in language teaching and claims that teaching writing skills by means of WhatsApp was more effective than through regular instruction. Moreover, it increased the learners' levels of motivation. The author states that the WhatsApp tool creates a positive social environment, encouraging a sense of belonging to a community or a team with other learners as well as the teacher. It also reduces anxiety. Therefore, the utilization of mobile devices in education should be seen as vital.

Andujar [26] claims that WhatsApp with its mobile instant messaging shows the potential to improve the student's writing skills in the second language and activate their involvement. In addition, WhatsApp seems to be accepted among students.

Khansarian-Dehkordi and Ameri-Golestan [21] examined the way mobile learning influences both acquisition and retention of vocabulary and concluded that even though the traditional method brings benefits, the results of those who used mobile phones or tablet PCs with a social networking application Line were significantly better, and these students themselves noticed their improvement. Nonetheless, the authors emphasize that technology cannot replace the physical classroom. Mobile devices should rather be used to encourage learners to interact with each other in the virtual world and create a fun environment for mutual learning.

Zhang [31] studied the effect of an app called English Fun Dubbing (EFD). The author claims that EFD supports the student's language learning autonomy by providing them with an opportunity to practice by themselves at their own pace. Zhang [31] concluded that a reasonable choice of a suitable application not only enhances learning but also makes students use mobile devices in more reasonable ways than they usually do.

According to Klimova [3], research indicates that mobile apps help develop all language skills, primarily retaining new vocabulary, and the use of these apps also increases students' motivation to study. Using the apps also boosts confidence, class participation, and students' tendency to use mobile devices in education [3].

Gamlo [24] emphasizes the importance of motivation to learn English. The author believes in using mobile game-based language learning. Nevertheless, the apps should be selected based on students' interests, needs and level.

Hwang et al. [28] conducted research into using video clips with either full captions, i.e., showing all the words in the same language as the audio output; partial captions, i.e., showing only key words in the same language; or partial captions with subtitles, i.e., key words in the language of audio output (captions) and their translations (subtitles) into the language of the students. Concerning learning motivation, the students learning with full captions showed significantly higher motivation than those learning with partial captions and subtitles. The authors also deem it important to differentiate between active-style and reflective-style students. The latter prefer learning by thinking to learning by interacting with videos.

Naderi and Akrami [29] investigated the effect of reading comprehension instruction by means of Telegram (Messenger) groups. According to these authors, online instruction has become popular, and their results suggest that students prefer the mobile phone as the best tool for reading short texts.

Similarly, Aghajani [27] looked into the influence of m-learning on cooperative learning (two or more people learning together) and compared face-to-face instruction of English writing with that by means of Telegram. They conclude that Telegram makes the learning environment more meaningful and it helps improve students' writing performance. In addition, Telegram, as the authors claim, actively encourages a cooperative environment and increases motivation.

Leis et al. [4] focus mainly on the effects that using smartphones in class has on students' autonomy, by which is meant their study outside the classroom. Their findings show that students encouraged to use smartphones in class tend to study more outside the classroom and are more autonomous learners than those who are restricted from using smartphones in class. Therefore, the authors strongly advise teachers to allow their students to use smartphones for language learning in class.

Hao et al. [22] studied how m-learning may benefit weak students of English as a foreign language. They contend that low-achieving students, often marginalized in class, may regain the sense of accomplishment with the help of effective mobile technology applications. Similarly, most other students also improve.

## 4. Discussion

On reading these selected articles, there is a feeling that not only language learning and teaching but education as such is on the threshold of a profound change. It may seem that the traditional model is on the wane (e.g., [23,30]). The use of mobile devices in education seems to be on an inevitable

rise. The problem lies in the way they should be used. Therefore, it is vital to determine potential advantages as well as drawbacks of m-learning utilization in education.

It seems to be true that the penetration of smartphones and the potential utilization of mobile devices make m-learning a great opportunity. Most young and adult learners use smartphones all the time. Teachers as well as their peers can approach them at almost any time. It may streamline communication. Mobile devices can be used for storing study materials, which significantly decreases heavy loads that children have to carry on their backs. Moreover, the Internet enables the learner to access target language content that they are interested in. For instance, if the student wants to study geography, there are a lot of texts available as well as video clips on the subject. At the same time, m-learning has become a major developing area in education. It is no coincidence that there are a lot of teachers who have started using m-learning in their classes and researchers who conduct research into it. Last but not least, the way that people live in the 21st century supports using mobile technologies in education.

On the other hand, some apps used by learners are not designed by language experts. Moreover, it seems that students should be guided and controlled in using language learning mobile apps for various reasons, including the lack of self-confidence in using new technologies or an unsuitable language level of apps used for particular students. Other downsides of using smartphones in education could be small screens, external interference as well as distraction, the addictive nature of smart devices, and the sometimes unfavorable attitudes of some teachers and parents.

Some questions also remain to be answered about m-learning (see Table 1), including the following. Should/will m-learning remain a supportive medium or become the primary one in education? Can we really expect profound changes in education, including a paradigm change? If so, how to best prepare for it? What new trends can be expected in m-learning? Is it better to create (a lot of) new apps or to utilize already existing platforms? What is the best way to guide students in m-learning? It may also be relevant to take into account whether the student prefer an active or reflective style of learning [28].

There are other review articles focusing on m-learning. The review article by Klimova [5] assessed 15 original articles and the findings include the following facts. First, mobile apps are effective in developing all skills, particularly vocabulary. Second, students' perceptions towards using mobile technologies for language learning are positive. Third, students using mobile technologies for language learning are more motivated to learn both inside and outside class. Klimova [5] also listed both benefits and limitations linked to using m-learning in language learning.

Another review by Hwang and Fu [13] had a wider scope and studied 93 papers, dividing their research into two periods, i.e., 2007–2011 and 2012–2016. It uncovers the following trends. First, most mobile-assisted language learning teaches English as a foreign/second language. Second, researchers' attention was paid mainly to higher-education students, whereas pre-school children had rarely been the subject of such studies. Third, research on vocabulary was the most common. Fourth, higher order thinking, e.g., problem solving, critical thinking, creativity, and communicative competence, only became an important issue between 2012 and 2016, whereas prior to 2012 it had not appeared in the reviewed articles at all. According to Hwang and Fu [13], earlier studies on m-learning usually focused on teaching individual language skills, whereas nowadays it is more common to deliver multiple language skills in authentic learning environments. The authors also maintain that most studies agree on the effectiveness of m-learning.

Thus, the answer to the research question is positive since the findings of this review study reveal that there is a potential in the use of mobile apps. Moreover, the use of apps contributes to the enhancement of the learner's cognitive capacity, the learner's motivation to study in both formal and informal settings, the learner's autonomy and confidence, as well as the promotion of personalized learning, helping low-achieving students to reach their study goals. However, to achieve the effectiveness of these apps, it is desirable to design, plan and implement them with caution, according to students' needs, and to deliver multiple language skills in authentic learning environments.

**Table 1.** Mobile learning (m-learning) SWOT analysis.

| SWOT Analysis: m-Learning | | | |
|---|---|---|---|
| **Strengths** | Mobile apps effectively develop all language skills. | **Weaknesses** | Cautious design, planning and implementation is sometimes missing but desirable. |
| | Students embrace using mobile technologies for language learning. | | Respect to students' needs. |
| | Students are more motivated to study. | | Essential to deliver multiple language skills in authentic learning environments. |
| | M-learning is becoming a salient feature of education. | | Small screen size of mobile devices. |
| | Enhancement of the learner's cognitive capacity. | | Lack of human contact. |
| | The learner's increasing autonomy and growing confidence. | | External interference, distraction. |
| | More personalized learning. | | The addictive nature of mobile devices. |
| | Diversified resources. | | Technical problems. |
| **Opportunities** | A lot of potential in m-learning as a new trend. | **Threats** | It is not clear whether m-learning should remain a supportive medium or become the primary one in education |
| | The fast development of Web 2.0, 3.0, X.0. | | Difficult to assess if profound changes in education should be expected, including a paradigm change: if so, how to best prepare for these changes? |
| | The rapid development of mobile and smart technologies. | | Chaotic environment—a lot of new apps of varying quality plus the utilization of already existing platforms. |
| | May make full inclusion in education possible. | | Potential lack of guidance for students in m-learning environments. |
| | New learning environment. | | Potential problems for students preferring a reflective style of learning to an active one. |

Source: authors' own processing.

## 5. Conclusions

The results reveal that m-learning is becoming a salient feature of education as it is a great opportunity and an immense step forward, and it should be supported especially thanks to the benefits it brings for language learning. These include: the enhancement of the learner's cognitive capacity, the learner's motivation to study in both formal and informal settings, the learner's autonomy and confidence, as well as the fact that it promotes personalized learning and helps low-achieving students to reach their study goals. Although it seems to be effective overall, it is desirable to design, plan and implement m-learning with caution, according to students' needs, and to deliver multiple language skills in authentic learning environments.

The limitations of this review consist in the different methodologies conducted in the detected studies, as well as different subject samples (varying from only 10 [22] to 140 [4]) and the researching of different language skills. Future research should focus on the effectiveness of the use of such mobile apps for teaching all four language skills in the context of the learner's performance.

**Author Contributions:** Conceptualization, J.K., B.K.; Methodology, B.K.; Software, not applicable; Validation, B.K.; Formal analysis, J.K.; Investigation, J.K., B.K.; Resources, B.K.; Data curation, J.K.; Writing—original draft preparation, J.K., B.K.; Writing—review & editing, J.K.; Visualization, N/A; Supervision, B.K.; Project administration, N/A; Funding acquisition, N/A.

**Funding:** This research received no external funding.

**Acknowledgments:** This paper was supported by the research project SPEV 2104/2019, run at the Faculty of Informatics and Management, University of Hradec Kralove, Czech Republic. The authors thank Aleš Berger for his help with data collection.

**Conflicts of Interest:** The authors declare no conflict of interest.

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
