# Peer review of "Use of Smartphone Applications in English Language Learning—A Challenge for Foreign Language Education"

_education, doi:10.3390/educsci9030179_

Round 1
Reviewer 1 Report
Very solid research. The path of their implementation is presented in the logical figure 1. The SWOT analysis deserves recognition - a very interesting and transparent data compilation. Conclusions short, understandable and most importantly convincing for further research - "Future research should focus on the effectiveness of the use of such mobile apps teaching all four language skills in context on learner’s performance."
Author Response
Thank you.
Reviewer 2 Report
Review for education-546078
The abstract needs rewriting for english grammar nuances - it currently reads slightly disjointed and doesn't quite flow easily as a narrative.
The Research Question is very broad, however as this is a literature review and the potential start of a research project that is ok.
The results and discussion sections would be enhanced with more explicit structuring around the themes identified, or an explicit indication that the results section summarises each of the 16 articles with either date or author order.
The SWOT analysis is useful, and the paper will be helpful to others in the domain of CALL/MALL.
Overall a well written paper that deserves publication after some revisions.
Author Response
The abstract needs rewriting for english grammar nuances - it currently reads slightly disjointed and doesn't quite flow easily as a narrative. – The abstract has been modified with the help of a native speaker. Please see the main manuscript.
The Research Question is very broad, however as this is a literature review and the potential start of a research project that is ok. - OK
The results and discussion sections would be enhanced with more explicit structuring around the themes identified, or an explicit indication that the results section summarises each of the 16 articles with either date or author order. – The sentence has been added to the part on Results: The key findings of the selected studies are described in more detail below (p. 4, line 137).
The SWOT analysis is useful, and the paper will be helpful to others in the domain of CALL/MALL. – Thank you.
Overall a well written paper that deserves publication after some revisions.
Reviewer 3 Report
The article has a good structure and it is well written. I suggest to extend theoretical background. The research question is not very well worded, as it is a qualitative study, it should be asking 'what, why, how' to answer with a full answer. However, the answer is missing. In the analyses, it would be useful to add what students are meant in those research studies (line 125, 138, etc.), as it is not the same for all age levels. Discussion should include the findings of own research and comparisons with other studies. This discussion only covers other review studies. Line 222 states that "most young learners use smartphones all the time". I believe it is not only young learners who use smartphones all the time, and this review study is probably also not covering only young learners. As it was mentioned before, there is no answer to the research question.
Author Response
The article has a good structure and it is well written. I suggest to extend theoretical background. – Some information has been added into the Introduction. Please see the main manuscript. (p. 2, lines 53-55, 59-61).
The research question is not very well worded, as it is a qualitative study, it should be asking 'what, why, how' to answer with a full answer. However, the answer is missing. – The research question has been specified and the answer is stated in the part on Discussion (p. 7, lines 272-278).
In the analyses, it would be useful to add what students are meant in those research studies (line 125, 138, etc.), as it is not the same for all age levels. – This has been explained (p. 4, lines 142-143).
Discussion should include the findings of own research and comparisons with other studies. – This has been done since the research of one of the authors is compared with other studies (e.g., [3, 5, 12]). This discussion only covers other review studies.
Line 222 states that "most young learners use smartphones all the time". I believe it is not only young learners who use smartphones all the time, and this review study is probably also not covering only young learners. – Yes, we agree. This statement has been modified (p. 6, line 229).
As it was mentioned before, there is no answer to the research question. – This has been done as stated above.